# The Impact of Direct Oral Anticoagulant Prophylaxis for Thromboembolism in Thrombophilic Patients Undergoing Abdominoplastic Surgery

**DOI:** 10.3390/healthcare10030476

**Published:** 2022-03-03

**Authors:** Pasquale Verolino, Caterina Sagnelli, Roberto Grella, Giovanni Francesco Nicoletti, Antonello Sica, Mario Faenza

**Affiliations:** 1Multidisciplinary Department of Medical Surgical and Dental Specialties, Unit of Plastic Surgery, University of Campania “Luigi Vanvitelli”, 80120 Naples, Italy; pasqualeverolino@libero.it (P.V.); info@robertogrella.it (R.G.); giovannifrancesco.nicoletti@unicampania.it (G.F.N.); mariofaenza@gmail.com (M.F.); 2Department of Mental Health and Public Medicine, University of Campania “Luigi Vanvitelli”, 80131 Naples, Italy; 3Department of Precision Medicine, University of Campania “Luigi Vanvitelli”, 80131 Naples, Italy; antonello.sica@fastwebnet.it

**Keywords:** abdominoplasty, thrombophilia, thromboprophylaxis, venous, thromboembolism, hematoma

## Abstract

Congenital or acquired thrombophilia is observed in 10–15% of the general population; therefore, careful screening is carried out in patients at higher risk of venous thrombo-embolism (VTE). High risk of VTE is a contraindication in patients undergoing abdominoplasty. We evaluated rivaroxaban, an oral Xa inhibitor, with enoxaparin, a subcutaneously low molecular weight heparin (LMWH), in 48 female patients with documented thrombophilia, undergoing thrombo-prophylaxis after abdominoplasty. Patients were stratified into two groups according to thrombo-prophylaxis procedure: enoxaparin Group (*n* = 28) and rivaroxaban Group (*n* = 20). Hematologic outcomes were evaluated including VTE and hematoma. No episodes of VTE occurred in both groups; two patients during their course of enoxaparin presented severe hematoma for drainage and hemostasis revision. This study suggests that abdominoplasty, in patients with thrombophilia, in combination with thrombo-prophylaxis can be performed safely. Rivaroxaban was as effective as LMWH for preventing VTE, with only a moderate risk of clinically relevant bleeding. More research is needed to determine the optimal timing and duration of prophylaxis in patients undergoing plastic surgery.

## 1. Introduction

Thrombophilia describes a heterogeneous group of hereditary and/or acquired conditions that predispose individuals to venous or arterial thromboembolic events. Arterial thrombosis may occur after the erosion or rupture of an atherosclerotic plaque and, through platelet-mediated thrombi, can cause ischemic injuries especially in tissues with a terminal vascular bed. Myocardial infarction and ischemic stroke are among the most frequent arterial thrombotic events that occur in thrombophilic states, much rarer events are visceral and retinal ischemia [1,2].

Venous thromboembolism (VTE) has an incidence of 1.69–1.98 per 1000 in the general population, including annual deep vein thrombosis and pulmonary embolism with an incidence of 1.24 and 0.6 per 1000 per year, respectively [3,4].

Congenital or acquired thrombophilia is present in about 10–15% of the general population. The most important thrombophilic deficits, such as protein C, protein S and ATIII deficiency, have a 15–30 times higher thrombosis rate compared to the general population, with a prevalence of 0.14–0.5% for C protein, 0.1–1% for S protein and 0.02–0.17% for TIII deficit.

These clinical conditions can be further split based on the affected mechanism in abnormal function and defects and disorders of coagulation factors. The most common forms of thrombophilia are due to inherited abnormal function of coagulation factors are Factor V Leiden and the G2021A prothrombin. Factor V Leiden affects about 5–15% of the general population. The G2021A prothrombin autosomal dominant mutation is present in 1% to 5% of the general population and leads to elevate circulating plasma levels of prothrombin.

Venous thrombosis is a multifactorial event and gene environment interaction is important. Risk factors for VTE can be subdivided into factors that promote venous stasis, factors that promote blood hypercoagulability, and factors causing endothelial injury or inflammation. In terms of patient related risk factors has been demonstrated how an over 30 body mass index (BMI), age older than 50 years, female sex, smoking habit, use of oral contraceptives, hormone replacement therapy and a thrombophilic state could lead to a significant increase in VTE risk in surgical setting [5,6]. In addition, these different categories of thrombogenic risk factors may interact synergistically to further increase the overall risk of VTE, and, furthermore, the relative risks of VTE increase dramatically in combination with acquired hypercoagulability factors [7]. Estimates of the relative risk of VTE with different thrombophilias are varied and are uncertain.

The relative risk for VTE is increased approximately 3- to 8-fold in heterozygous carriers of FVL and 18 to 80-fold in homozygous carriers [8,9,10].

The relative risk for VTE is increased approximately 2- to 5-fold in heterozygous carriers of the G20210A prothrombin polymorphism while homozygous carriers have a clearly higher risk, although the magnitude is not well defined. Coinheritance of both FVL and G2021A prothrombin heterozygous mutations occurs in 1:1000 in the general population and is associated with a 7- to-20-fold increase in the relative risk of venous thrombosis [11,12,13,14,15].

VTE is a common complication of surgical procedures; the risk for VTE is determined by combination of individual predisposing factors and the specific type of surgery [16]. In patients undergoing general surgery without anti-thrombotic prophylaxis, the rate of deep vein thrombosis was 15–30% and fatal pulmonary embolism 0.2–0.9% [16].

Risk factors for VTE in general surgery patients include cancer as the reason for surgery, duration of procedure, previous VTE, advanced age and obesity [17]. However, it is appropriate to perform a thrombophilia screening for patients who have highest VTE risk, based on careful clinical history [14,18]. Moreover, the use of appropriate prophylactic measures against VTE in all surgical procedures is recommended [19].

In plastic surgery field abdominoplasty procedures have reported the highest rate of deep vein thrombosis, estimated between 1% and 9.4% [20,21]. The reason why these procedures have such a high risk of VTE should be related with both to the characteristics of patients and procedures. In abdominoplasty setting, operative procedures consist in wide dissection, damage to the superficial veins, long operative time, decreased venous return after plication of the diastasis recti, increased intrabdominal pressure and minimal postoperative mobilization [22,23]. Therefore, the high risk of VTE is among the relative contraindications in patients undergoing abdominoplasty, due to the highest occurrence among aesthetic procedures [5,24,25]. For that reason, two major approaches have been used to evaluate and stratify the risk of VTE in surgical patients, one focused on the patient and the other on the surgical procedure. The procedure-based approach is the one suggested by Bates et al. [26], whereas Caprini et al. [27] is the major supporter of the patient based VTE risk assessment. From 2001 all the patient-based risk assessment models in VTE prevention have been progressively merged into a one single entity, known as the Caprini Risk Assessment Score, that is based on an aggregated risk score calculated on 39 single risk factors [28,29].

In 2015 Pannucci et al. [30] in plastic surgery setting demonstrated that anticoagulant prophylaxis should be administered in patients showing a Caprini score greater than eight to reduce the risk of VTE [28]. In patients with a Caprini score of 7 or more, surgeons should strongly consider subcutaneous administration of low molecular weight heparin (LMWH) as antithrombotic prophylaxis [31,32].

Rivaroxaban is an oral factor Xa inhibitor FDA-approved in 2011 for VTE prophylaxis in patients undergoing orthopedic surgical procedure, such as hip or knee replacement. Rivaroxaban is an orally active inhibitor of factor Xa used with increasing frequency in venous thromboembolism treatment and prevention and stroke prophylaxis in atrial fibrillation. Common side effects are nausea, diarrhea and constipation. Its effects can last for up to 3 days after the last dose and some delayed spontaneous bleeding [33,34,35] and major bleeding in patients with active cancer have been described [36].

Hunstad et al. [37]—in a retrospective chart review, from September 2012 to July 2014, of 132 patients undergoing abdominoplasty surgery and who received rivaroxaban postoperatively—observed deep vein thrombosis/pulmonary embolism in one patient (0.76%). In the same abdominoplasty procedure, other authors observed an incidence of 1.1% for deep vein thrombosis and 0.8% for pulmonary embolism in [38,39].

In this paper, we compare the use of rivaroxaban and enoxaparin in thrombo-prophylaxis after abdominoplasty procedure in a cluster of patients affected by inherited thrombophilia, or a personal history of VTE, or a familial history of VTE undergoing abdominoplasty. The aim of this study was to assess thrombotic complications following sclerotherapy in thrombophilic patients, comparing thromboprophylaxis with LMWH or direct oral anticoagulants (DOACs).

## 2. Materials and Methods

A retrospective cohort study analyzing 48 thrombophilic female patients who undergoing abdominoplasty procedure was conducted over a 5-year period, from September 2014 to July 2019 at Multidisciplinary Department of Medical Surgical and Dental Specialties, Plastic Surgery Unit, University of Campania “Luigi Vanvitelli”, Naples, Italy.

Thrombophilia was defined as either the diagnosis of an inherited thrombophilia, a personal history of VTE or a familial history of VTE [21,40,41,42,43].

Patient demographics characteristics, biochemical tests, comorbidities, operative details, and post-operative complications were collected. Kidney and liver function tests were performed, and triglyceride and cholesterol levels, and coagulation were determined according to routine methods at the baseline of abdominoplastic surgery and during the follow-up period.

The body mass index (BMI; kg/m^2^) and waist circumference were determined by standard procedures at the baseline of abdominoplastic surgery and during the follow-up period.

In according to the procedure of our Plastic Surgery Unit in all patients at pre-operatory evaluation are compiled the Caprini Risk Assessment Score. In all 48 patients, the Caprini Risk Assessment Score for VTE pre-operative was made and all the 48 patients showed a score of 7 or more, for that reason antithrombotic prophylaxis was administered.

To obtain a better clinical and surgical outcome, and a betted compliance of patients, the choice on the way of administration of anti-thrombotic prophylaxis drug was made in according to the patient’s inclination. Patients were then divided into two groups based on the drugs employed in anti-thrombotic prophylaxis:−Rivaroxaban Group: rivaroxaban 10 mg oral 6–8 h post-operatively and then once a day for 7 days post-operatively.−Enoxaparin Group: enoxaparin 4000 UI subcutaneously 8–12 h pre-operatively and then once a day for 7 days post-operatively.

We collected data for all groups of patients in terms of drain permanence, thromboembolic events such as hematoma, deep vein thrombosis and pulmonary embolism, and on the other surgical complication.

Thromboembolic events are defined as the presence of a venous thrombus verified with color Doppler ultrasonography (lower limbs for all patients with high Caprini Risk Score Assessment, and for symptomatic patients Doppler arterial upper limbs and CT angiography are used). The hematoma is defined as the formation of fluid collection of superficial blood nature outside of blood vessels, and clinically evaluated.

The hemorrhagic complications correlated with the VTE prophylaxis drug, such as nosebleeds, menorrhagia, gingival bleeding, etc. were also evaluated.

All subjects were observed for a follow-up of at least three months after surgery with physician’ visits at 15 days, 3 weeks and two and three months after surgery.

Once discharged, patients were instructed to contact the Centre for any clinical event that may occur.

We described the characteristics of our study population using counts and percentages for categorical variables. The results are reported as frequencies, and the data for continuous variables are presented as mean values (M) ± standard deviation (SD). Student’s *t*-test, the Mann–Whitney and the χ^2^ test were used to compare continuous or categorical variables. A *p* value level < 0.05 was considered statistically significant.

## 3. Results

Table 1 shows the characteristics of the patients: age, BMI, smoking habit, the Caprini Score, drain permanence and complications (Table 1).

A total of 14 patients out of 48 (29.16%) were affected by heterozygosis of Factor V Leiden. A total of 8 patients out of 48 (16.66%) were affected by G2021A prothrombin autosomal dominant mutation. A total of 10 patients out of 48 (20.83%) reported personal history of VTE. A total of 16 patients out of 48 (33.33) reported familiar history of VTE.

In the enoxaparin Group, we observed 28 patients treated with enoxaparin does of 4000 UI subcutaneously 8–12 h preoperatively and then once a day for 7 days postoperatively such as anti-thrombotic prophylaxis, and in the rivaroxaban Group we enrolled 20 patients treated with rivaroxaban does of 10 mg oral 6–8 h postoperatively and once a day for 7 days postoperatively.

The surgical procedure was performed under epidural anesthesia and consisted in abdominoplasty with plication of diastasis recti. Operative time was 90 to 120 min.

Two suction drains were placed, and external compression garments was applied for 30 days postoperatively.

Suction drains were removed once the drainage volume was less than 40 mL in 24 h.

Post-operative analgesia was administered with a 48-h infusion pump preloaded with paracetamol and ketorolac.

Mobilization of patients was performed 8 to 12 h post-operatively and all the patients were discharged from the ward 48 h post-operatively.

No statistically significant differences were found in patients characteristics (Table 1).

We did not find statistically significant difference between the two groups both in terms of post-operative thromboembolic events (*p* = 0.992), drain permanence (*p* = 0.842) or hematoma (*p* = 0.680). Only two cases of hematoma were observed in the enoxaparin Group developed only in the paraumbilical site. In addition, appearance of hemorrhagic complications such as nosebleeds, menorrhagia, gingival bleeding, etc. was not observed in both groups.

## 4. Discussion

Thrombophilia is defined as a primary or secondary clinical condition that predisposes to thrombosis. It is important to underline that a patient affected by thrombophilia does not necessarily requires pharmacological prophylaxis or treatment, but a patient affected by thrombophilia is at high risk for post-operative VTE [44].

VTE is the third most common cause of vascular mortality worldwide and is strongly related both to the type of surgical procedure and to patients’ characteristics [45]. Among reconstructive plastic surgery procedures, abdominoplasty in combination or not with other body contouring procedures presents the highest risk of VTE with an incidence rate between 1% and 9.4% as well as an incidence rate of one in 1502 cases [20,46]. Clinically symptomatic VTE occurs with high frequency after post-bariatric body contouring surgery, particularly in circumferential abdominoplasty (7.7%), abdominoplasty (5.0%) and less frequently in upper body and in the breast remodeling (2.9%) [47].

Despite no relative risk data are provided to scientifically support, the Caprini score has been universally accepted to evaluate the VET risk factor, by the most important surgical association [48,49]. The Caprini scoring system is conceptually based on four levels of risk factors of VTE as following “low risk” (0–1 points), “moderate risk” (2 points), “high risk” (3–4 points) and “highest risk” (≥5 points) [50]. The Caprini scoring system has been validated for 30-day VTE events in patients undergoing general, urological, and vascular surgery [51]. Moreover, modifications to the model were validated in post-bariatric body contouring patients [52] and hospitalized patients [53,54]. Several studies reported the good reliability of the Caprini scoring system. Pannucci et al. [32], between March 2006 and June 2009, validated the Caprini scoring system risk assessment model in 1126 plastic and reconstructive surgery patients, accepting it as the gold standard of patient care. Pannucci et al. reported one in nine patients (11.3%) with a Caprini score more than 8 had a VTE event within 60 days of surgery and among patients with a Caprini score of 7–8 or more than 8 there was no evidence that the risk of VTE was limited to the immediate post-operative period [32]. The Caprini scoring system is a useful and effective tool for stratifying patients in plastic and reconstructive surgery for the risk of VTE [32].

However, the prevention of complications, particularly severe complications such as VTE, is a high priority in plastic aesthetic surgery [55]. The ideal prophylaxis for prevention of VTE should have efficacy, low risk of side effects, reasonable cost and should be independent from patient weight or renal function, easily administrable and tolerable. Unfortunately, no pharmacological treatment can be defined as ideal as we always must deal with the risks and benefits of drug administration in each individual patient.

The timing of VTE prophylaxis is still widely debated. The standard dose for enoxaparin is 40 mg administered subcutaneously once a day and 30 mg subcutaneously in-patient with poor renal function.

The direct oral anticoagulants have been successfully employed in treatment of DVT in patients affected by thrombophilia [56,57,58,59,60,61,62,63,64]. Rivaroxaban can be orally administered for VTE prevention for hip or knee replacement surgery showing postoperative bleeding rates like enoxaparin [64,65,66,67]. The main advantage of direct oral anticoagulants, such as rivaroxaban, is their predictability in pharmacokinetic and pharmacodynamic profiles, not requiring a continuous laboratory monitoring and leading to a simplify dosing regimens well tolerated by patients [33]. Hunstad et al. [37], on 132 patients undergoing abdominoplasty, demonstrated the safety of oral administration of rivaroxaban for VTE, with low rates of symptomatic VTE and hematoma requiring surgical revision. Morales et al. [60] in a retrospective chart review from January 2012 to February 2015 on 1572 patients undergoing body contouring procedures observed how direct oral anticoagulants (rivaroxaban and apixaban) are comparable to low molecular weight heparin for VTE prophylaxis had similar rates in terms of drug-related complications. Instead, Swanson et al. [63] expressed perplexity about the use of direct oral anticoagulants in VTE prevention as the rates of bleeding complications are unacceptable in body contouring surgery.

We admit that the present study has some limitations. First, this is a single-centre retrospective study, and therefore, further study is needed for validating. In fact, many cofactors are implicated in determining VET as factors that promote venous stasis, factors that promote blood hypercoagulability, and factors causing endothelial injury or inflammation, some correlated to the subject and same to the specific surgery.

An important limitation of the study is the choice of treatment carried out not by randomization but according to the patient’s inclinations to ensure a better compliance to the therapy, carried out because both drugs were on the market. Another limitation is the small size of the sample studied, which however, for the results obtained, we think it may be sufficient to stimulate the setting up of large multicenter studies.

## 5. Conclusions

In this paper we conducted a cohort study on 48 consecutive abdominoplasties on a homogenous population of patients affected by thrombophilia or with history of VTE.

Patients were divided into two groups of 28 and 20 in which VTE prophylaxis was performed with enoxaparin and rivaroxaban, respectively.

The message that emerges from our study is that in this cluster of patients at high risk of developing VTE there is no there is no statistically significant difference in VTE prophylaxis between LMWH and direct oral anticoagulants, especially during abdominoplasty procedures both in terms of thromboembolic events and in terms of post-operative bleeding and drain permanence.

## 6. Patents

### Statement of Ethics

All procedures performed were in accordance with the international guidelines, with the Helsinki Declaration of 1975 (revised in 1983) and followed the Italian laws of privacy and with the local Ethics Committees named: “Comitato Etico Universita’ Degli Studi Della Campania “Luigi Vanvitelli”-Azienda Ospedaliera Universitaria “Luigi Vanvitelli”-Azienda Ospedaliera Rilievo Nazionale “Ospedali Dei Colli”, Naples, Italy. Each patient signed an anonymous informed consent for the use of his data for anonymous clinical investigations and scientific publications. At the baseline visit, each patient signed an informed consent to surgical procedure.

## Figures and Tables

**Table 1 healthcare-10-00476-t001:** Characteristics of the study population, according in antithrombotic prophylaxis.

Characteristics	Rivaroxaban Group *n* = 20 Patients	Enoxaparin Group *n* = 28 Patients	*p*-Value
Age, years, (Mean ± SD)	46.75 ± 10.08	48.67 ± 9.36	0.5144
BMI, kg/m^2^, (Mean ± SD)	27.10 ± 5.67	27.07 ± 4.80	0.9838
Smoking, N (%)	9 (45%)	12 (42.85%)	0.8827
Caprini Score (Mean ± SD)	8.55 ± 1.86	8.53 ± 0.77	0.9547
Drain Permanence, days (Mean ± SD)	7.15 ± 1.91	7.00 ± 0.745	0.842
Complications, *n* (%):			
Hematoma	0	2	0.680
Thromboembolic Events	0	0	0.992

BMI—Body mass index.

## Data Availability

Data sharing not applicable.

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
