# Peer review of "The Impact of Direct Oral Anticoagulant Prophylaxis for Thromboembolism in Thrombophilic Patients Undergoing Abdominoplastic Surgery"

_healthcare, 2022, doi:10.3390/healthcare10030476_

Round 1

Reviewer 1 Report

Verolino et al. Present a manuscript comparing the clinical thrombosis outcome in patients undergoing abdominoplasty and being on LMWH vs anti-Xa DOACs. Overall, this is nice work demonstrating the equivalence in efficacy between the LMWH and anti-Xa inhibitor in respect to thrombotic complications. Authors, however, need to address the following concerns: 

  1. Have authors followed on the bleeding complications like epistaxis, menorrhagia, gum bleedings, etc.?
  2. Authors mention there were 2 cases of hematoma in enoxaparin group. Please describe it further. Were hematomas developing paraumbilically? Or were they seen somewhere else? 
  3. Extensive English editing is required.
  4. Citations needed for sentences in line 36 and line 37. 
  5. In line 89 authors state that "thrombophilic states are rather venous than arterial". Although this may be true from a prevalence standpoint, the way this sentence is structured would mislead the reader. Revise accordingly, since there a number of arterial thromboses that thrombophilias can cause. 
  6. In line 87 - authors state "mild thrombophilia". Severity of thrombophilic states are now yet well categorized, therefore more clarification needs to be provided here. 
  7. Enoxaparin and rivaroxaban have to be lower case as they are not brand names. 
  8. Line 150 - it should be "anti-thrombotic" rather than "an-thrombotic". 
  9. P values have to presented in Table 1. 
  10. Line 179 - authors first mention 39 factors, and then in line 179 state 40. Which is true? 

Author Response

Reviewer(s)' Comments to Author:
Reviewer #1:

Verolino et al. Present a manuscript comparing the clinical thrombosis outcome in patients undergoing abdominoplasty and being on LMWH vs anti-Xa DOACs. Overall, this is nice work demonstrating the equivalence in efficacy between the LMWH and anti-Xa inhibitor in respect to thrombotic complications. Authors, however, need to address the following concerns: 

Point 1: Have authors followed on the bleeding complications like epistaxis, menorrhagia, gum bleedings, etc.?

Answer to the Reviewer point 1: The observation of the reviewer has been accepted and the manuscript was modified accordingly.

Point 2: Authors mention there were 2 cases of hematoma in enoxaparin group. Please describe it further. Were hematomas developing paraumbilically? Or were they seen somewhere else? 

Answer to the Reviewer point 2: The observation of the reviewer has been accepted and the manuscript was modified accordingly.

Point 3: Extensive English editing is required.

Answer to the Reviewer Point 3: The new manuscript has been evaluated by an expert of English language.

Point 4: Citations needed for sentences in line 36 and line 37. 

Answer to the Reviewer point 4: The observation of the reviewer has been accepted and the manuscript was modified accordingly.

Point 5: In line 89 authors state that "thrombophilic states are rather venous than arterial". Although this may be true from a prevalence standpoint, the way this sentence is structured would mislead the reader. Revise accordingly, since there a number of arterial thromboses that thrombophilias can cause. 

Answer to the Reviewer point 5: The observation of the reviewer has been accepted and the manuscript was modified accordingly.

Point 6: In line 87 - authors state "mild thrombophilia". Severity of thrombophilic states are now yet well categorized, therefore more clarification needs to be provided here. 

Answer to the Reviewer point 6: The observation of the reviewer has been accepted and the manuscript was modified accordingly.

Point 7: Enoxaparin and rivaroxaban have to be lower case as they are not brand names. 

Answer to the Reviewer point 7: The observation of the reviewer has been accepted and the manuscript was modified accordingly.

Point 8: Line 150 - it should be "anti-thrombotic" rather than "an-thrombotic". 

Answer to the Reviewer point 8: The observation of the reviewer has been accepted and the manuscript was modified accordingly.

Point 9: P values have to presented in Table 1. 

Answer to the Reviewer point 9: The observation of the reviewer has been accepted and the table 1 was modified accordingly.

Point 10: Line 179 - authors first mention 39 factors, and then in line 179 state 40. Which is true? 

Answer to the Reviewer point 10: The observation of the reviewer has been accepted and the manuscript was modified accordingly.

Reviewer 2 Report

The authors report experience from their surgical center with performing abdominoplasty in women with "thrombophilia" which is not clearly defined. The results are of interest to both surgeons and haematologists/haemostaseologists. However, I have several concerns which must be adequately addressed before the manuscript is suitable for publication.

1) In the Abstract you state that “two patients during their course of Rivaroxaban presented severe hematoma”, but in Table 1 it appears that the two hematomas occurred in the enoxaparin group? Please clarify.

2) The introduction is not well written. The structure is unclear, and the authors repeat themselves many times. For example, the section begins with introducing inherited thrombophilias and VTE risk in these conditions, then proceeds to surgery, then back again to thrombophilia where it repeats itself and even states other numbers for prevalences and VTE risks than previously stated. Furthermore, suitable references are lacking in some places. For the present study to be acceptable for publication, the introduction should be thoroughly revised.

Specific comments for the introduction:

A) On page 1, lines 37-43, you state that: “The most important thrombophilic deficits, such as protein C, protein S and ATIII deficiency, have a 15-30 times higher thrombosis rate in the general population, with a prevalence of 0.14%-0,5 for C protein, 0.1-1% for S protein and 0.02-0.17% for TIII deficit. In the feared Factor V Leiden and PT 20210 gene homozygous, that present an about 80 times increased thrombotic risk, considering the frequency among general population, the rarity of these cases is more evident.” Could you please cite relevant references for these prevalences and relative risks? Also, I know that relative risk estimates for VTE with different thrombophilias may vary and are uncertain, especially for the rarer thrombophilias. Nonetheless, an 80 times increased thrombosis risk with FV Leiden or FII 20210 homozygosity seems very high to me; it has been reported considerably lower e.g. in Sode et al CMAJ 2013 doi: 10.1503/cmaj.121636.

B) Page 2, lines 47-48: “…and they present a highest thrombotic risk, 2-3% for 20210 gene, 3-8% for Factor V Leiden [1].” This sentence is not clear. 2-3% of what for the 20210 polymorphism and 3-8% of what for FV Leiden? Also, “20210 gene” should be “20210 polymorphisms, or “G20210A polymorphism” to be completely correct.

C) Page 2, lines 87-88: “In this regard it is widely demonstrated that almost half of surgical patients who developed VTE have a mild thrombophilic syndrome [18].” This is rather an astonishing statement, and as your source, you cite a paper from the “Australian Family Physician”. Furthermore, what does “a mild thrombophilic syndrome” even mean? Please explain, or consider omitting this statement altogether.

3) In the Materials and methods section, several questions arise:

A) Page 3, lines 129-130: “We followed up a consecutive homogenous cohort of 48 thrombophilic female patients who underwent abdominoplasty from September 2014 to July 2019.” First, how did you define “thrombophilic”? Second, did you perform a retrospective or “historical” study, sitting down in 2019 to review all medical records 5 years back – or did you perform a prospective study, starting in 2014 and concluding in July 2019? This should be clearly stated.

B) Page 3, lines 131-133: “We compiled the Caprini Risk Assessment Score for VTE preoperatively and all the patients showed a score of 7 or more, for that reason antithrombotic chemoprophylaxis was administered.” Do you routinely perform a Caprini assessment for all patients in your clinic?

C) Page 3, lines 133-134: “Patients were then divided into two groups according to the drugs employed in antithrombotic chemoprophylaxis:” How was it decided which patients received which form of prophylaxis? This is a very important point and should be described clearly.

D) Page 3, lines 140-141: “We collected data for all groups of patients in terms of drain permanence, thromboembolic events, and hematoma formation.” How did you define and verify a thromboembolic event? How did you define and verify excessive hematoma formation? How long was the follow-up time? And how did you obtain this information?

E) Page 3, line 142: The Mann-Whitney test was used for what?

4) The Discussion section could benefit from a language revision, especially in the latter part, to make the flow better. Also, a brief discussion on the limitations of your study should be included.

Author Response

Reviewer(s)' Comments to Author:

Reviewer 2:

(x) Extensive editing of English language and style required

Answer to the Reviewer: The new manuscript has been evaluated by an expert of English language.

The authors report experience from their surgical center with performing abdominoplasty in women with "thrombophilia" which is not clearly defined. The results are of interest to both surgeons and haematologists/haemostaseologists. However, I have several concerns which must be adequately addressed before the manuscript is suitable for publication.

Point 1: In the Abstract you state that “two patients during their course of Rivaroxaban presented severe hematoma”, but in Table 1 it appears that the two hematomas occurred in the enoxaparin group? Please clarify.

Answer to the Reviewer point 1:  we are very sorry for the mistake, and the manuscript was modified accordingly.

Point 2: The introduction is not well written. The structure is unclear, and the authors repeat themselves many times. For example, the section begins with introducing inherited thrombophilias and VTE risk in these conditions, then proceeds to surgery, then back again to thrombophilia where it repeats itself and even states other numbers for prevalences and VTE risks than previously stated. Furthermore, suitable references are lacking in some places. For the present study to be acceptable for publication, the introduction should be thoroughly revised.

Answer to the Reviewer point 2: The criticism of the reviewer has been accepted and the manuscript was modified accordingly.

Point 3: Specific comments for the introduction:

  1. On page 1, lines 37-43, you state that: “The most important thrombophilic deficits, such as protein C, protein S and ATIII deficiency, have a 15-30 times higher thrombosis rate in the general population, with a prevalence of 0.14%-0,5 for C protein, 0.1-1% for S protein and 0.02-0.17% for TIII deficit. In the feared Factor V Leiden and PT 20210 gene homozygous, that present an about 80 times increased thrombotic risk, considering the frequency among general population, the rarity of these cases is more evident.” Could you please cite relevant references for these prevalences and relative risks? Also, I know that relative risk estimates for VTE with different thrombophilias may vary and are uncertain, especially for the rarer thrombophilias.

Nonetheless, an 80 times increased thrombosis risk with FV Leiden or FII 20210 homozygosity seems very high to me; it has been reported considerably lower e.g. in Sode et al CMAJ 2013 doi: 10.1503/cmaj.121636.

  1. B) Page 2, lines 47-48: “…and they present a highest thrombotic risk, 2-3% for 20210 gene, 3-8% for Factor V Leiden [1].” This sentence is not clear. 2-3% of what for the 20210 polymorphism and 3-8% of what for FV Leiden? Also, “20210 gene” should be “20210 polymorphisms, or “G20210A polymorphism” to be completely correct.
  2. C) Page 2, lines 87-88: “In this regard it is widely demonstrated that almost half of surgical patients who developed VTE have a mild thrombophilic syndrome [18].” This is rather an astonishing statement, and as your source, you cite a paper from the “Australian Family Physician”. Furthermore, what does “a mild thrombophilic syndrome” even mean? Please explain, or consider omitting this statement altogether.

Answer to the Reviewer point 3: The observation of the reviewer has been accepted and the manuscript was modified accordingly.

Point 4: In the Materials and methods section, several questions arise:

  1. A) Page 3, lines 129-130: “We followed up a consecutive homogenous cohort of 48 thrombophilic female patients who underwent abdominoplasty from September 2014 to July 2019.” First, how did you define “thrombophilic”? Second, did you perform a retrospective or “historical” study, sitting down in 2019 to review all medical records 5 years back – or did you perform a prospective study, starting in 2014 and concluding in July 2019? This should be clearly stated.
  2. B) Page 3, lines 131-133: “We compiled the Caprini Risk Assessment Score for VTE preoperatively and all the patients showed a score of 7 or more, for that reason antithrombotic chemoprophylaxis was administered.” Do you routinely perform a Caprini assessment for all patients in your clinic?
  3. C) Page 3, lines 133-134: “Patients were then divided into two groups according to the drugs employed in antithrombotic chemoprophylaxis:” How was it decided which patients received which form of prophylaxis? This is a very important point and should be described clearly.
  4. D) Page 3, lines 140-141: “We collected data for all groups of patients in terms of drain permanence, thromboembolic events, and hematoma formation.” How did you define and verify a thromboembolic event? How did you define and verify excessive hematoma formation? How long was the follow-up time? And how did you obtain this information?
  5. E) Page 3, line 142: The Mann-Whitney test was used for what?

Answer to the Reviewer point 4: The observation of the reviewer has been accepted and the manuscript was modified accordingly.

Point 5: The Discussion section could benefit from a language revision, especially in the latter part, to Answer to the Reviewer point 5: The observation of the reviewer has been accepted and the manuscript was modified accordingly.

We thank the Editor and the Reviewers for helping us to improve our paper.

The manuscript has been read and approved by all the authors.

We also declare that we have no conflict of interest in connection with this paper.

We sincerely hope that the enclosed manuscript can be accepted for publication in the: Healthcare

Prof.ssa Caterina Sagnelli

Round 2

Reviewer 2 Report

The manuscript has been greatly improved. I have one major comment. It is still not clear how thrombophilia was defined. You write (page 3, lines 129-130): "Only women with thrombophilia defined as a factor correlated with thrombotic phenomena linked to congenital or acquired factors. [21,40-43]"

I suggest you alter the sentence, e.g. to something like: "Thrombophilia was defined as either the diagnosis of an inherited thrombophilia, a personal history of VTE or a familial history of VTE." If that is how you defined thrombophilia?

Also, another round of language editing may be suitable.

Author Response

To the Editor in Chief of Healthcare

We re-submitted our article “The impact of direct oral anticoagulant prophylaxis for thromboembolism in thrombofilic patients underwent abdominoplastic surgery”, Manuscript ID: healthcare-1600106, Section: Environmental Factors and Global Health, Special issue: Skin Disorders in Hematological Disease.

The following changes (shown underlined). The manuscript has been improved according to the suggestions of the reviewer:

Reviewer(s)' Comments to Author:
Reviewer 2:

Point 1: The manuscript has been greatly improved. I have one major comment. It is still not clear how thrombophilia was defined. You write (page 3, lines 129-130): "Only women with thrombophilia defined as a factor correlated with thrombotic phenomena linked to congenital or acquired factors. [21,40-43]"

I suggest you alter the sentence, e.g. to something like: "Thrombophilia was defined as either the diagnosis of an inherited thrombophilia, a personal history of VTE or a familial history of VTE." If that is how you defined thrombophilia?

Answer to the Reviewer Point 1: The observation of the reviewer has been accepted and the table 1 was modified accordingly.

Point 2: Also, another round of language editing may be suitable.

Answer to the Reviewer Point 2: The new manuscript has been evaluated by an expert of English language.

We thank the Editor and the Reviewers for helping us to improve our paper.

The manuscript has been read and approved by all the authors.

We also declare that we have no conflict of interest in connection with this paper.

We sincerely hope that the enclosed manuscript can be accepted for publication in the: Healthcare.

Prof.ssa Caterina Sagnelli
